# Vehicle Surface Pressure Prediction from 2D Sketches via a Pre-Trained Diffusion Model

**Shunya Nakamura**
Honda Motor Co., Ltd.
Saitama, Japan
`shunya_nakamura@jp.honda`

**Osamu Ito**[*]
Honda Motor Co., Ltd.
Saitama, Japan
`osamu_ito@jp.honda`

## Abstract

In the early stages of automotive design, designers explore shape concepts using 2D sketches, yet existing aerodynamic evaluation methods require 3D geometry representations. We propose an end-to-end approach that directly predicts surface pressure coefficient (Cp) distributions from 2D sketch images via image-to-image translation. By fine-tuning a pre-trained image editing diffusion model (Qwen-Image-Edit-2511) with LoRA on the large-scale automotive aerodynamics dataset DrivAerNet++, we achieve Relative L2 Error (Rel L2) = 0.165 and $R^2$ = 0.955 on the test set. Even with training data reduced to 2.2% (128 samples), the model maintains Rel L2 = 0.218, demonstrating applicability in practical settings where CFD simulation costs are prohibitive. We confirm good generalization to unseen base vehicle categories while revealing that prediction quality degrades for shape features absent from training data, highlighting that coverage of shape features matters more than category labels. We also investigate calibrated uncertainty (UQ) via diffusion ensemble, revealing when variance reliably indicates prediction error. These results demonstrate the feasibility of aerodynamic evaluation without 3D geometry representations, opening a path toward aerodynamic feedback at the earliest design stage.

## 1 Introduction

In automotive design, both aesthetics and aerodynamic performance are critical factors affecting product competitiveness. Improvements in aesthetics have been reported to substantially increase sales (Burnap et al., 2021), while aerodynamic characteristics strongly influence fuel efficiency and driving stability. These two aspects often involve trade-offs, making early-stage evaluation and adjustment crucial in the design process.

The automotive design workflow typically progresses through four stages: Concept, Feasibility, Pre-Production, and Production. Aerodynamic evaluation employs different methods at each stage, but all existing approaches assume the availability of 3D geometry. Prior to the Concept stage, designers explore shape ideas through 2D sketches, yet no means exist to evaluate surface pressure distributions at this earliest phase since 3D models have not been created. Consequently, aerodynamically unfavorable trends may only become apparent from the Concept stage onward, necessitating iterative shape modifications and re-evaluations. If aerodynamic trends could be assessed at the sketch stage, designers could make aerodynamically informed decisions when shape flexibility is greatest.

Recently, data-driven aerodynamic prediction has offered approaches to address this challenge.

**First**, approaches that predict directly from 3D geometry exist. CarBench (Elrefaie et al., 2025b) establishes a benchmark for predicting surface pressure from 3D point clouds, evaluating geometric deep learning methods including PointNet (Qi et al., 2017), Point Transformer (Zhao et al., 2021), and AB-UPT (Alkin et al., 2025). However, these methods require 3D geometry representations and thus cannot be applied at the sketch stage.

**Second**, approaches that route through 3D geometry from 2D input exist. Design Agents (Elrefaie et al., 2025a) proposes a framework connecting sketches to aerodynamic evaluation via DeepSDF-

---

[*]Corresponding author.

based 3D shape retrieval and generation. While promising in eliminating manual 3D CAD creation and enabling evaluation within minutes from sketches, the 3D model generation involves either retrieving the most similar existing shape from the DrivAerNet++ (Elrefaie et al., 2024) database or generating/interpolating shapes using DeepSDF's latent space. Consequently, the evaluated 3D geometry is not a precise trace of the sketch lines but rather a geometrically coherent shape deemed most similar within the learned latent space or database. This constraint leaves the risk that the evaluated shape diverges from the designer's intended geometry.

**Third**, approaches that directly predict physical quantities from 2D input without intermediate 3D geometry are conceivable. For aerodynamic prediction from 2D images, Song et al. (2023) represented car shapes using multi-view 2D images and achieved drag coefficient prediction ($R^2 > 0.84$) with CNN surrogates, though the output was limited to a single scalar value. For physics prediction using diffusion models, DiffFluid (Luo et al., 2024) formulated flow field prediction as image-to-image translation, demonstrating that diffusion models are effective for physical state estimation. Research incorporating physical laws into diffusion processes includes Bastek et al. (2025), who introduced PDE residuals as loss terms, and Huang et al. (2024), who addressed PDE solution recovery from partial observations. However, these existing studies focus primarily on 2D fluid fields and have not explored application to surface pressure distributions of 3D external flows around vehicle bodies.

The data-driven methods described above all assume large-scale training data comprising thousands of samples, yet high-fidelity CFD simulations are computationally expensive per case, making construction of such datasets impractical in many real design settings. Against this constraint, leveraging pre-trained models offers a promising approach.

Building on the third approach above, this work proposes an end-to-end framework for directly predicting surface pressure coefficient (Cp) distributions from 2D vehicle sketches.

Specifically, we take multi-view line drawing sketches as input and adapt the pre-trained image editing model Qwen-Image-Edit (Wu et al., 2025) with LoRA (Hu et al., 2021) to predict Cp distributions via image-to-image translation. Since diffusion models are stochastic generative processes where outputs vary for identical inputs, we apply diffusion ensemble (averaging multiple samples) at inference, as proposed by Shu & Farimani (2024), to improve accuracy. Furthermore, we investigate calibrated uncertainty (UQ) by examining whether ensemble variance reliably indicates prediction error, and discuss implications for dataset construction. Training and evaluation use the DrivAerNet++ dataset.

The main contributions of this work are:

1. **Demonstrating feasibility of end-to-end Cp prediction**: We show that LoRA adaptation of a pre-trained diffusion model enables high-accuracy prediction of 3D surface pressure distributions from 2D sketches.

2. **Improving prediction accuracy via diffusion ensemble**: We confirm that averaging predictions across multiple seeds improves accuracy compared to single predictions.

3. **Verifying data efficiency**: We progressively reduce training data and confirm that practical accuracy is maintained even with limited data.

4. **Analyzing generalization characteristics**: We reveal that coverage of shape features, rather than category labels, governs generalization performance.

## 2 METHOD

### 2.1 PROBLEM FORMULATION

We learn a mapping $f : \mathcal{X} \to \mathcal{Y}$, where $\mathcal{X}$ is the space of 2D sketch images and $\mathcal{Y}$ is the space of surface pressure coefficient distributions represented as 2D images. Given a sketch image $x \in \mathbb{R}^{H \times W \times 3}$ that integrates six views of a vehicle into a single image, we predict the corresponding surface pressure coefficient (Cp) distribution $y \in \mathbb{R}^{H \times W \times 3}$. Image resolution is standardized to 1024×1024 pixels. Cp is a dimensionless quantity normalizing local pressure by dynamic pressure and is standard in aerodynamic analysis (see Appendix A for the definition). The six views (front, back, top, bottom, left, right) are arranged in a fixed 3×2 grid layout, consistent across all samples. All views use parallel projection with a shared fixed focal point and camera distance, so that vehicle size differences are preserved at a consistent scale. This fixed layout is used because full vehicle geometry information is required to determine the flow field. Pressure values are encoded to RGB using the jet colormap, with Cp values clipped to $[-1, 1]$ (see Appendix B for details).

### 2.2 DATASET

We use DrivAerNet++, a large-scale automotive aerodynamics dataset containing 8,129 vehicle designs with CFD-computed surface pressure distributions. Note that all Cp values in this dataset are computed under fixed CFD conditions (Reynolds number, turbulence model, boundary conditions); predictions from our model are therefore specific to these settings. The dataset comprises three base categories—Fastback (F), Estateback (E), and Notchback (N)—combined with underbody configurations to form eight subcategories (see Appendix C for details). For input images, we use sketches rather than 3D renderings to simulate early design stage conditions. Sketches are binary line drawings (black lines on white background) generated from 3D meshes via normal maps and image-to-image translation, eliminating style variation and retaining only shape information (see Appendix D for generation procedure details). Target images are Cp values from 3D meshes rendered as 2D images from the same viewpoints. Figure 1 shows an example of training data.

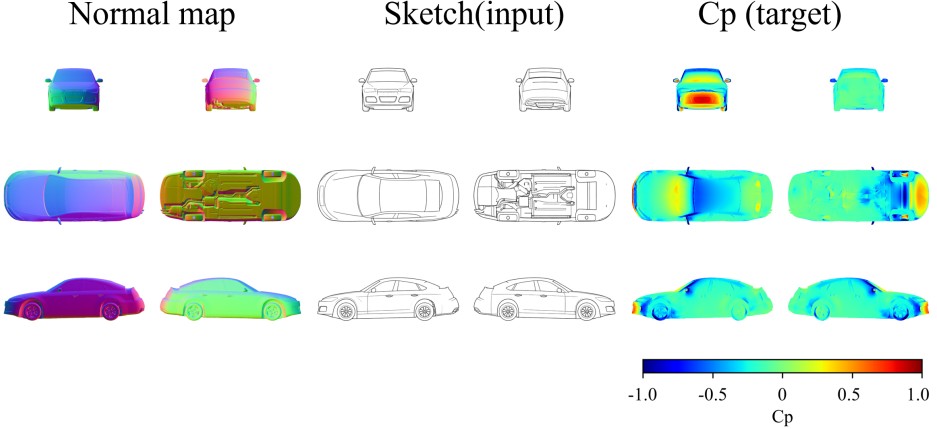

Figure 1: Training data example. Left: Normal map. Center: Sketch image (input). Right: Cp distribution (target).

### 2.3 MODEL

We adopt Qwen-Image-Edit-2511 (Wu et al., 2025; Qwen Team, 2025) as the base model for Cp prediction. This is an image editing model based on MM-DiT (Multimodal Diffusion Transformer) with approximately 20B parameters, featuring a dual-encoder design (combining Vision-Language Model semantic embeddings with VAE visual latent representations) that enables transformation following text instructions while preserving input image structure. The model was developed with consideration for engineering applications including industrial design, featuring precise image editing and image drift mitigation. These characteristics suit our task of generating pressure distribution

colormaps while maintaining sketch contour structure. We fine-tune this model with LoRA (hyperparameters in Section 3). Text prompts to the model are uniformly set to "run CFD" to avoid prompt content influencing predictions.

## 2.4 EVALUATION METRICS

We use Rel L2 as the primary metric, following CarBench, the standard benchmark for automotive aerodynamics prediction, and DiffFluid, which demonstrated diffusion model application to fluid prediction. This metric normalizes prediction error by the ground truth norm, enabling fair comparison across samples. During evaluation, both predicted and ground truth images are decoded to Cp values via inverse jet colormap transformation, and errors are computed in Cp value space. Note that direct comparison with prior work is not appropriate due to differing input/output modalities, but using the same metric allows sharing evaluation criteria (see Appendix A for definition). As auxiliary metrics, we report the coefficient of determination ($R^2$) and the mean absolute error (MAE). $R^2$ provides intuitive interpretation of prediction accuracy, while MAE indicates error magnitude in Cp value scale.

## 3 EXPERIMENTS

### 3.1 EXPERIMENTAL OVERVIEW

**Dataset and splits.** Experiments use the DrivAerNet++ dataset (Section 2.2). We follow the splits defined in the original paper (Elrefaie et al., 2024) (train: 5,817, validation: 1,148, test: 1,154), which guarantee no shape overlap between splits.

**Experimental structure.** We conduct experiments from three perspectives:

1. **Baseline establishment (Section 3.2)**: Verify feasibility of the proposed task and optimize inference parameters and LoRA settings. Since optimal settings for applying diffusion models to physics prediction tasks are unknown, we conduct systematic exploration.

2. **Data efficiency (Section 3.3)**: Verify data efficiency enabled by combining pre-trained models with LoRA. Since CFD simulations are computationally expensive, learning capability with limited data is practically important.

3. **Cross-category generalization (Section 3.4)**: Evaluate generalization to vehicle categories not included in training. As CarBench notes, practical value of surrogate models heavily depends on generalization ability to unknown shapes, since designers often explore new shapes not included in training data.

### 3.2 BASELINE ESTABLISHMENT

**Factors affecting performance.** In image translation with diffusion models + LoRA, major factors affecting performance are categorized as: (1) training parameters (rank, alpha, training steps), (2) inference parameters (steps, CFG scale). Additionally, diffusion ensemble (averaging multiple samples) as an inference strategy also affects performance. Since these factors can interact, exhaustive exploration of all combinations is computationally prohibitive.

**Experimental order design.** We proceed in the following logical order. First, we confirm task feasibility and training convergence using official recommended settings (Section 3.2.1), verifying preconditions for subsequent experiments. Next, using the trained model, we optimize inference parameters and diffusion ensemble (Section 3.2.2), which can be efficiently explored without additional training. We then conduct sensitivity analysis of LoRA hyperparameters (Rank, Alpha) (Section 3.2.3), validating settings used in subsequent experiments. Finally, we compare subset evaluation with full evaluation to verify evaluation methodology validity and report final baseline performance (Section 3.2.4).

### 3.2.1 FEASIBILITY VERIFICATION

**Aim.** Confirm that the proposed task of "predicting pressure distributions from sketches" is feasible.

**Settings.** We use the Qwen-Image-Edit official recommendations of 40 inference steps and CFG=4.0. LoRA uses rank=32, alpha=16 ($\alpha/r$=0.5), trained with AdamW (learning rate $1 \times 10^{-4}$) using all training data (5,817 images). Training used NVIDIA RTX 6000 Ada ×2 (approximately 36 hours for ~29,000 steps). Validation used 8 samples, one from each subcategory.

**Results.** Figure 2 shows the training curve. The curve shows rapid improvement in early stages (~6,000 steps), with convergence tendency confirmed after approximately 15,000 steps. Training for approximately 29,000 steps achieved Rel L2 = 0.197, $R^2$ = 0.937, with no signs of overfitting. These results confirm feasibility of the proposed task.

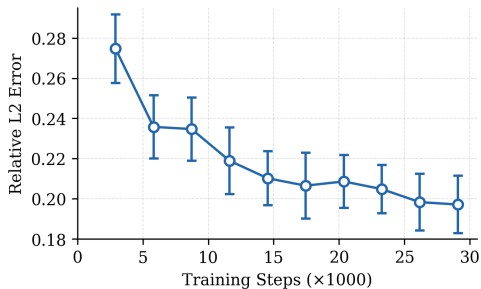

Figure 2: Training convergence curve. Rel L2 vs. training steps.

Figure 3 shows qualitative results for a representative sample.

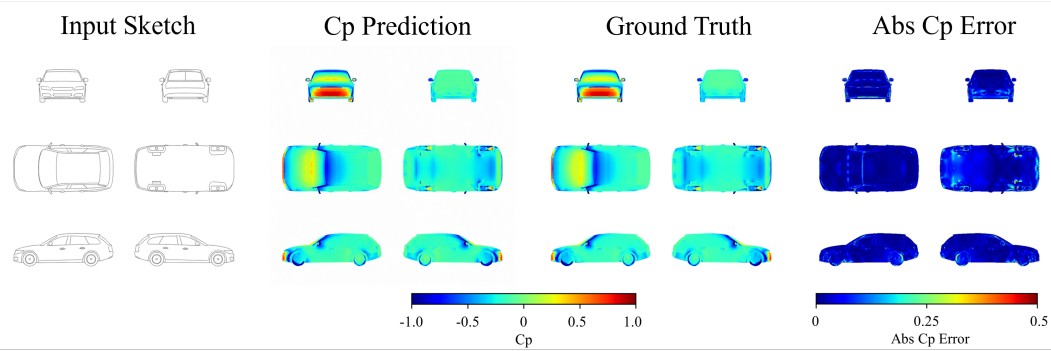

Figure 3: Qualitative results for sample E_S_WWC_243 (Estateback). Predictions accurately capture global pressure distribution patterns. Results for all 8 samples are shown in Appendix E.

### 3.2.2 INFERENCE PARAMETER OPTIMIZATION

**Aim.** Verify effects of inference parameters (diffusion steps, CFG scale) and diffusion ensemble.

**Settings.** We use the model trained in Section 3.2.1 (Rank=32, Alpha=16, ~29,000 steps).

For inference parameters: (1) with CFG=4.0 fixed, we explore steps $\in \{2, 3, ..., 10, 30\}$; (2) with steps=3 fixed, we explore CFG $\in \{1.0, 4.0, 7.0, 10.0\}$.

For diffusion ensemble, we evaluate sample counts $\in \{1, 2, ..., 20\}$.

Each evaluation uses 200 samples extracted via stratified random sampling from the validation set ($n$=1,148). This sample size provides approximately 6–7% relative margin of error (95% CI) around the estimated mean, which is sufficient precision for comparing conditions.

**Results.** Figure 4 shows inference parameter optimization results.

**Diffusion steps** (Figure 4 left): Counter-intuitively, fewer steps yielded better results. steps=3 was best (Rel L2 = 0.176), outperforming steps=30 (Rel L2 = 0.190).

**CFG scale** (Figure 4 center): Lower values showed better performance. CFG=1.0 was best (Rel L2 = 0.170), with small difference from CFG=4.0 (Rel L2 = 0.176). Clear performance degradation occurred at CFG≥7.0. Since input sketch images already provide strong structural information, additional conditioning via CFG may have become excessive. We adopt the commonly used CFG=4.0 for reproducibility.

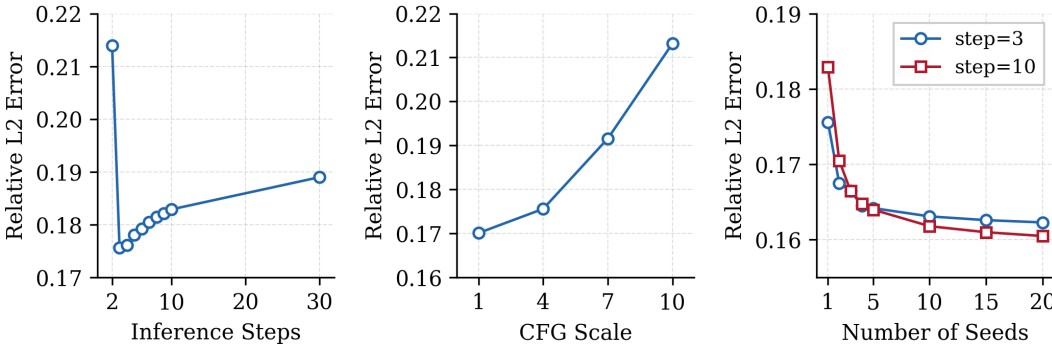

Figure 4: Inference parameter optimization results. Left: Effect of step count. Center: Effect of CFG scale. Right: Effect of diffusion ensemble. Error bars indicate the standard deviation of Rel L2 across the 8 evaluation samples.

Table 1: Ensemble variance: statistics of mean Rel L2 across 20 random seeds ($n$=200 validation samples per seed)

| Steps | Mean Rel L2 | Std. Dev. | Range |
|-------|-------------|-----------|-------|
| 3     | 0.177       | 0.001     | 0.005 |
| 10    | 0.188       | 0.003     | 0.012 |

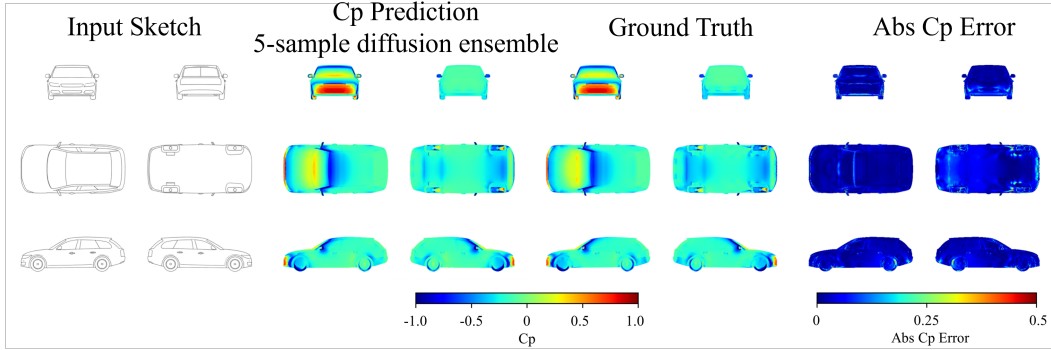

Figure 5: Qualitative effect of diffusion ensemble. Ensemble averaging reduces noise and improves prediction quality.

**Diffusion ensemble** (Figure 4 right): Performance improved from single diffusion sample (Rel L2 = 0.176) to 5-sample ensemble (Rel L2 = 0.164), with minimal improvement beyond 5 samples. Figure 5 shows qualitative effects of diffusion ensemble (detailed comparison with individual samples shown in Appendix F).

**Relationship between steps and ensemble**: Comparing steps=3 and steps=10, steps=3 is superior for single samples, but a reversal occurs where steps=10 becomes slightly superior with 20-sample ensemble. This phenomenon can be interpreted from a bias-variance tradeoff perspective. Table 1 shows ensemble variance. steps=3 has standard deviation 0.001 while steps=10 has 0.003; steps=3's variance is smaller. steps=3 maintains prediction stability by strongly preserving input image structure (low variance) while having systematic deviation (bias) due to coarse discretization. In contrast, steps=10 is more susceptible to initial noise causing prediction variance (high variance) but generates predictions closer to true values on average (low bias) through sufficient denoising steps. Based on standard MSE decomposition $MSE = bias^2 + \sigma^2/n$, when sample count $n$ is small, the variance term dominates making low-variance steps=3 advantageous, while as $n$ increases, the variance term diminishes making low-bias steps=10 advantageous. At 5 samples, both achieve equivalent performance, with minimal additional improvement beyond 5 samples. Additionally, steps=3 is $3.3\times$ faster than steps=10.

**Inference settings for subsequent experiments.** Based on these results, subsequent experiments use steps=3, CFG=4.0, 5-sample ensemble. Although CFG=1.0 was best, the difference from 4.0 was small, so we select the widely-used 4.0 for reproducibility. Training curves are evaluated with single diffusion samples. The low-variance characteristics of steps=3 (standard deviation 0.001) enable statistically reliable estimation even with single samples. Note that diffusion ensemble mean and variance calculations are performed in Cp value space after decoding each sample output via inverse jet transformation.

### 3.2.3 LoRA Hyperparameter Sensitivity Analysis

**Aim.** Analyze how LoRA Rank and Alpha affect prediction performance and validate hyperparameters used in subsequent experiments.

**Settings.** We conduct grid search over Rank $\in \{32, 64, 128\}$ and Alpha $\in \{16, 32, 64, 128\}$ combinations. All conditions use 5,817 training images and train for approximately 29,000 steps (equivalent to 10 epochs). Training curves are evaluated using 200 samples from the validation set with single diffusion samples.

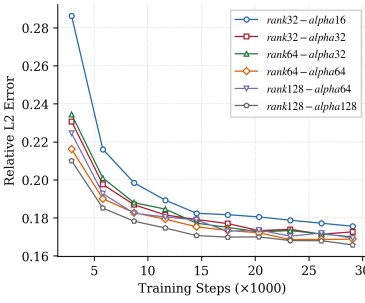

Figure 6: Training curves for each LoRA Rank $\times$ Alpha combination.

Table 2: LoRA Rank $\times$ Alpha grid search results (5-sample ensemble)

| Rank | Alpha | $\alpha/r$ | Rel L2 | $R^2$ | MAE |
|------|-------|-----------|--------|-------|------|
| 32 | 16 | 0.5 | 0.166 | 0.955 | 0.0253 |
| 32 | 32 | 1.0 | 0.163 | 0.957 | 0.0243 |
| 64 | 32 | 0.5 | 0.160 | 0.958 | 0.0237 |
| 64 | 64 | 1.0 | 0.158 | 0.960 | 0.0230 |
| 128 | 64 | 0.5 | 0.159 | 0.960 | 0.0233 |
| 128 | 128 | 1.0 | 0.155 | 0.961 | 0.0225 |

**Results.** Table 2 and Figure 6 show performance at approximately 29,000 steps. Results confirm that performance improves with increasing Rank, but with pronounced diminishing returns. While some improvement is seen from Rank=32 to Rank=64, increasing from Rank=64 to Rank=128 shows near plateau, indicating saturation of additional parameter effects at high Rank. Moreover, all settings achieve $R^2 > 0.95$, showing that even the minimum configuration of Rank=32 provides sufficient expressiveness for this task. This suggests that the pre-trained model has already acquired rich visual representations, and relatively few additional parameters suffice for adaptation to fluid features. Based on the above, we judge Rank=32 to have sufficient expressiveness for this task and adopt Rank=32, Alpha=16 as the standard setting for subsequent experiments (Sections 3.3, 3.4) from computational efficiency considerations. This setting still achieves practical performance of Rel L2 = 0.166. Final evaluation on the test set is reported in Section 3.2.4.

### 3.2.4 SUBSET EVALUATION VALIDITY VERIFICATION

**Aim.** Sections 3.2.2 and 3.2.3 used 200-sample subsets for computational efficiency. This section verifies subset evaluation validity through comparison with full evaluation and reports final baseline performance.

**Settings.** Using the model trained with Rank=32, Alpha=16 for approximately 29,000 steps, we evaluate the full validation set ($n$=1,148) and full test set ($n$=1,154). Inference settings are steps=3, CFG=4.0, with single diffusion samples for validation and 5-sample diffusion ensemble for test. Inference takes approximately 13 seconds per sample on a single NVIDIA RTX 6000 Ada (approximately 67 seconds with 5-sample ensemble).

Table 3: Comparison of subset and full evaluation

| Eval target | $n$ | Rel L2 | $R^2$ | MAE |
|---|---|---|---|---|
| Val subset | 200 | 0.176 | 0.950 | 0.0260 |
| Val full | 1,148 | 0.176 | 0.950 | 0.0261 |
| Test subset | 200 | 0.166 | 0.955 | 0.0253 |
| Test full | 1,154 | 0.165 | 0.955 | 0.0252 |

**Results.** Table 3 shows results. For both validation and test sets, differences between subset and full evaluation are minimal, confirming that stratified sampling-based subset evaluation appropriately estimated population performance. Final baseline performance is Rel L2 = 0.165, $R^2$ = 0.955 on the full test set.

### 3.3 DATA EFFICIENCY

**Aim.** Verify data efficiency enabled by combining pre-trained models with LoRA. We progressively reduce training data and evaluate performance impact.

**Settings.** Training data is progressively reduced from 5,817 to 2,176, 1,024, and 128 images. To maintain category balance, each subset is created via stratified sampling preserving proportions across the eight vehicle configuration patterns. From computational cost considerations, approximately 15,000 steps (equivalent to 5 epochs on full dataset) is used as the training target. Evaluation uses 200 samples from the test set with 5-sample diffusion ensemble.

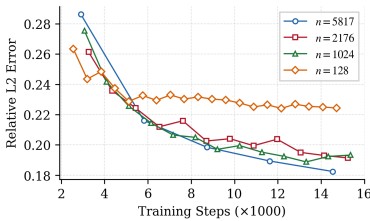

Figure 7: Training curves for each training size.

Table 4: Training size vs. prediction performance.

| Samples | Fraction | Rel L2 | $R^2$ | MAE |
|---|---|---|---|---|
| 5,817 | 100% | 0.171 | 0.953 | 0.0262 |
| 2,176 | 37.4% | 0.180 | 0.943 | 0.0286 |
| 1,024 | 17.6% | 0.182 | 0.946 | 0.0284 |
| 128 | 2.2% | 0.218 | 0.923 | 0.0352 |

**Results.** Figure 7 and Table 4 show performance at approximately 15,000 steps. Using full data (5,817 samples) Rel L2 = 0.171 as baseline, 2,176 samples (37%) achieves 0.180 and 1,024 samples (17.6%) achieves 0.182, showing gradual performance degradation against data reduction. Performance at 1,024 samples is nearly equivalent to 2,176 samples, suggesting redundancy in training data. Reducing to 128 samples (2.2%) yields Rel L2 = 0.218 with quantitative performance degradation, but qualitatively, global patterns are accurately captured (see Appendix G), at a level adequate for trend assessment in early design stages. These results demonstrate that leveraging pre-trained models enables efficient learning even with limited domain-specific data.

### 3.4 CROSS-CATEGORY GENERALIZATION

As described in Section 2.2, DrivAerNet++ comprises three base categories (F, E, N) combined with underbody configuration variations. CarBench provides comprehensive analysis of cross-category generalization; this section offers additional analysis. In the following, "E+N" denotes the combination of Estateback and Notchback, and "F+E+N" denotes all categories.

We first analyze in detail the performance degradation reported by CarBench when training on E+N and evaluating on Fastback, from the perspective of underbody configuration (Section 3.4.1). Based on those results, we verify base category generalization under limited data conditions (see Appendix I).

### 3.4.1 EFFECT OF UNDERBODY CONFIGURATION

**Aim.** Analyze in detail the performance degradation reported by CarBench when training on E+N and evaluating on Fastback.

**Settings.** Using a model trained on E+N, we evaluate Fastback separated by underbody configuration (Smooth/Detailed). Stratified extraction from the dataset follows Section 3.3, with training data size aligned to 2,176 samples.

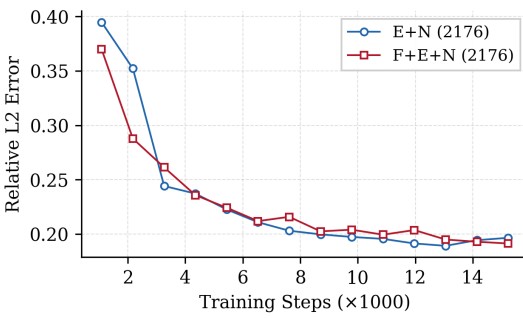

Figure 8: Training curves by category (validated on F_S).

Table 5: Cross-category prediction by underbody

| Training | Test | Rel L2 | $R^2$ | MAE |
|----------|------|--------|-------|-----|
| E+N | F_S | 0.186 | 0.943 | 0.030 |
| E+N | F_D | 0.410 | 0.856 | 0.066 |
| All | All | 0.180 | 0.943 | 0.029 |

**Results.** Figure 8 shows training curves for F+E+N (2,176 samples) vs. E+N only (2,176 samples), validated on F_S. As described in Section 2.2, Detailed underbody exists only in Fastback. Table 5 shows results evaluating Fastback with the E+N-trained model, separating F_D (Detailed underbody) and F_S (Smooth underbody).

For F_S, almost no performance degradation was observed, while significant degradation was confirmed for F_D. Prediction performance for F_S achieves the same level as when reducing training data including all categories to 2,176 samples in Section 3.3. This demonstrates that good generalization to unseen base categories is possible when underbody configuration matches. Meanwhile, the significant performance degradation for F_D suggests that cross-category generalization difficulty stems not from base category differences but from underbody configuration absent from training data. Qualitative results are shown in Appendix H.

From these results, we confirm that the model shows good generalization performance to unseen base categories. This is likely because parametric morphing included diverse outer body shapes in training data, covering shape information required for prediction. However, for shape features entirely absent from training data like Detailed underbody, prediction is difficult since required information is not provided. Additional verification of base category generalization under limited data conditions (128 samples or fewer) is shown in Appendix I.

## 4 DISCUSSION

To utilize aerodynamic prediction in early design stages, not only prediction values but also confidence information about "which regions' predictions should be trusted" is important. Diffusion ensemble (averaging predictions across multiple seeds) shown in Section 3.2.2 contributes to accuracy improvement, but ensemble variance (prediction spread) may simultaneously function as a confidence indicator. This section analyzes correspondence between high-variance regions and high-error regions.

Shu & Farimani (2024) reported strong correlation ($r \approx 1.0$) between ensemble variance and prediction error in PDE regression tasks. However, their study targeted PDE-specific models and continuous physical fields, differing in assumptions from this work. This work handles indirect mapping from 2D sketches to 3D surface pressure, where correspondence between input and output is more uncertain, making it inappropriate to expect equivalent correlation.

In this work, we analyze correlation between standard deviation across 5 seeds (seed=0–4) in Cp value space and absolute error between mean prediction and ground truth. Specifically, we compute correlation coefficients between standard deviation and absolute error for all pixels within the vehicle body region for each sample, reporting the average across the full test set. Table 6 shows results.

Table 6: Correlation between ensemble variance and prediction error (per-pixel, averaged over test samples)

| Training | Test | Mean Pixel Std (Cp) | Mean Pixel Err (Cp) | $r$ | Rel L2 |
|---|---|---|---|---|---|
| F+E+N (5,817) | F_D | 0.0102 | 0.0245 | 0.488±0.021 | 0.157 |
| E+N (2,176) | F_D | 0.0136 | 0.0653 | 0.299±0.036 | 0.410 |
| F+E+N (5,817) | F_S | 0.0098 | 0.0278 | 0.498±0.029 | 0.169 |
| E+N (2,176) | F_S | 0.0093 | 0.0296 | 0.466±0.032 | 0.186 |

Under in-distribution conditions (F+E+N training), moderate correlation of $r \approx 0.49$ was observed. When predicting F_D with E+N training, correlation coefficient dropped to 0.299. This indicates that correspondence between variance and error breaks down for shape patterns absent from training data. In contrast, correlation remained at 0.466 for F_S even with E+N training.

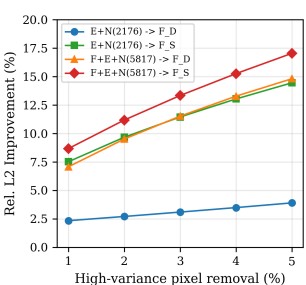

Figure 9: Rel L2 improvement vs. high-variance pixel exclusion rate. Legend: X→Y denotes training on X and testing on Y.

To quantitatively verify this correspondence, we analyzed Rel L2 improvement rates when excluding high-variance pixels. Figure 9 shows results. For F+E+N training and E+N training with F_S test, removing 5% of high-variance pixels yielded 14–17% improvement. In contrast, improvement for E+N training evaluated on F_D remained at approx. 4%. This indicates that for shapes absent from training data, the model "doesn't know what it doesn't know," where variance does not function as an error predictor. Qualitative visualization is shown in Appendix K.

This yields implications for diffusion ensemble uncertainty estimation. Under in-distribution conditions, ensemble variance is useful as a confidence indicator. Under out-of-distribution conditions, low variance should not be interpreted as "accurate prediction." Monitoring correlation coefficients and filtering improvement rates may identify coverage gaps in training data.

## 5 CONCLUSION

This work proposed an end-to-end framework for directly predicting surface pressure coefficient (Cp) distributions of 3D vehicles from 2D sketch images, demonstrating its feasibility. By adapting a pre-trained image editing diffusion model (Qwen-Image-Edit) with LoRA, we achieved prediction accuracy of Rel L2 = 0.165 and $R^2$ = 0.955 on the DrivAerNet++ dataset. This demonstrates that aerodynamic prediction is possible within 2D image space without routing through 3D point clouds or intermediate 3D geometry representations.

Key findings include: combining pre-trained models with LoRA achieves high data efficiency, maintaining Rel L2 = 0.218 even when reducing training data to 2.2% (128 samples). For generalization characteristics, good performance was shown for unseen base categories (Fastback), while significant degradation occurred for underbody configurations (Detailed underbody) entirely absent from training data, revealing that coverage of shape features matters more than category labels. For practicality, hand-drawn sketches achieved Rel L2 = 0.219, demonstrating that models trained on AI-generated sketches can handle actual design inputs (see Appendix J). For diffusion ensemble, beyond improving prediction accuracy, we discovered that correlation coefficients between ensemble variance and prediction error reflect training data coverage, suggesting potential for prioritizing data collection.

Two future research directions emerge. First, we will extend the framework to scalar quantities that designers ultimately need, such as the drag coefficient (Cd). Second, we will investigate application to wind-tunnel measurements. Discrepancy between CFD and wind tunnel tests is a practical issue; adapting this method to wind tunnel conditions using pressure tap measurement data as supervision could enable construction of surrogates consistent with wind tunnel conditions.

These results demonstrate feasibility of aerodynamic evaluation without requiring 3D geometry representations. Even at early design stages where 3D CAD does not exist, designers can potentially assess aerodynamic trends from sketches and make aerodynamically informed decisions when shape flexibility is greatest.

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

## A    NOTATION AND DEFINITIONS

This section defines the primary quantities used throughout the paper.

**Pressure Coefficient (Cp)**

The surface pressure coefficient (Cp) is a dimensionless quantity normalizing local pressure by dynamic pressure:

$$C_p = \frac{p - p_\infty}{\frac{1}{2}\rho_\infty V_\infty^2} \tag{1}$$

where:

- $p$: Local static pressure (pressure at each point on the surface)
- $p_\infty$: Freestream static pressure (reference pressure far from the body)
- $\rho_\infty$: Freestream density
- $V_\infty$: Freestream velocity
- $\frac{1}{2}\rho_\infty V_\infty^2$: Dynamic pressure (corresponding to kinetic energy)

Physical interpretation of Cp:

- **Cp = 1**: Stagnation point (flow completely stops, kinetic energy converts to pressure)
- **Cp = 0**: Same pressure as freestream
- **Cp < 0**: Lower pressure than freestream (region where flow accelerates)

**Evaluation Metrics**

We define the evaluation metrics used in this work.

**Rel L2 Error (Primary Metric)**

The primary metric used in CarBench and DiffFluid:

$$L_2^{\text{rel}} = \frac{\|y - \hat{y}\|_2}{\|y\|_2} \tag{2}$$

where $y$ is the ground truth and $\hat{y}$ is the prediction. Lower values indicate better predictions.

**Coefficient of Determination ($R^2$)**

Metric indicating correlation between predicted and ground truth values:

$$R^2 = 1 - \frac{\sum_i (y_i - \hat{y}_i)^2}{\sum_i (y_i - \bar{y})^2} \tag{3}$$

where $\bar{y}$ is the mean of ground truth values. Values closer to 1 indicate better predictions.

**Mean Absolute Error (MAE)**

Mean of absolute prediction errors:

$$\text{MAE} = \frac{1}{n}\sum_{i=1}^{n} |y_i - \hat{y}_i| \tag{4}$$

## B    Cp Encoding and Normalization

This section describes the encoding/decoding procedure for Cp values and verifies the validity of clipping to $[-1, 1]$.

**Encoding (Cp → RGB).** We use the matplotlib jet colormap to encode Cp values as RGB images. Cp values are first clipped to $[-1, 1]$, then linearly mapped to the colormap index range $[0, 1]$ via $(C_p + 1)/2$. This maps $C_p = -1$ to blue, $C_p = 0$ to cyan/green, and $C_p = 1$ to red.

**Decoding (RGB → Cp).** For evaluation, we decode predicted RGB images back to Cp values using an inverse lookup table. Specifically, we precompute the jet colormap at 256 discrete levels, then for each predicted RGB pixel, find the nearest neighbor in RGB space and retrieve the corresponding Cp value. This discretization introduces quantization error of at most $\Delta C_p = 2/256 \approx 0.008$, which is negligible compared to prediction errors (MAE $\approx 0.025$).

**Clipping Range Validation.** We now verify the validity of clipping Cp values to $[-1, 1]$ from both dataset analysis and design practice perspectives.

**Dataset Statistical Analysis.** We analyzed the Cp distribution across the entire DrivAerNet++ dataset (8,129 vehicles, approximately 500,000 surface mesh points per vehicle). Table 7 shows the statistics.

Table 7: Cp value statistics for DrivAerNet++ dataset

| Statistic | Value |
|---|---|
| Full dataset range | $[-51.7, 12.9]$ |
| Vehicle mean range | $[-3.69, 0.92]$ |
| All mesh points mean | $-0.18$ |
| All mesh points std. dev. | $0.30$ |
| $3\sigma$ range | $[-0.9, 0.9]$ |

The majority of Cp values fall within the $[-1, 1]$ range, and the $3\sigma$ range being $[-0.9, 0.9]$ indicates that $[-1, 1]$ clipping is consistent with the statistical distribution of the data.

**Spatial Distribution of Out-of-Range Regions.** Investigation of the spatial distribution of regions where $|\mathrm{Cp}| > 1$ confirmed that these extreme values are localized to specific regions of the vehicle body. High-pressure regions (Cp > 1) appear as numerical artifacts at sharp edges such as side mirror tips, while low-pressure regions (Cp < −1) are localized to tire surfaces, side mirror tips, and A-pillars. Figure 10 shows visualization of the vehicle with the most extreme Cp values in the dataset.

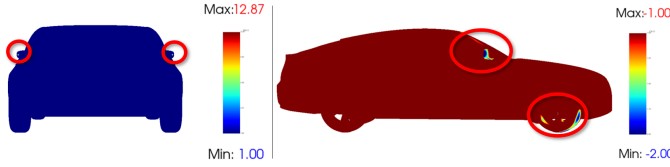

Figure 10: Vehicle with most extreme Cp values in dataset (min=−51.7, max=12.9). Only regions marked with red circles are outside the $[-1, 1]$ range, localized to side mirror tips, tires, and A-pillars.

**Limitations and Considerations of Clipping.** We consider the impact of information loss due to clipping. In conventional CFD analysis, local pressure extrema can influence the wake flow field, so clipping extrema may lose information on the "cause" side of physical causality. However, in our method, we clip CFD-computed results (the complete flow field including effects of extrema on the wake) as training data. That is, the training data retains the "effect" side of causality (Cp distribution across the entire body), potentially limiting the impact of extreme value clipping on

prediction performance. However, quantitative verification of this hypothesis remains for future work.

**Consistency with Design Practice.** In automotive aerodynamics design practice, visualization with $Cp \in [-1, 1]$ is standard, and what designers focus on in early stages are global pressure distribution trends arising from body shape. Our method's output range directly aligns with this practical convention.

**Conclusion.** From the above analysis, clipping to $[-1, 1]$ is valid in three aspects: (1) it covers the majority ($3\sigma$ range) of the dataset's Cp distribution, (2) out-of-range values are limited to accessories like tires and side mirrors, or sharp edges prone to numerical artifacts, and (3) it matches standard visualization ranges in design practice.

## C    DRIVAERNET++ VEHICLE CONFIGURATION PATTERNS

This section details the vehicle configuration patterns in the DrivAerNet++ dataset. The dataset comprises eight subcategories based on combinations of body type, underbody configuration, wheel type, tire type, and mirror presence.

Table 8: Vehicle configuration patterns in DrivAerNet++

| Label | Body | Underbody | Wheel | Tire | Mirror | $n$ |
|---|---|---|---|---|---|---|
| E_S_WW_WM | Estateback | Smooth | Open | Detailed | Yes | 698 |
| E_S_WWC_WM | Estateback | Smooth | Closed | Smooth | Yes | 688 |
| F_D_WM_WW | Fastback | Detailed | Open | Detailed | Yes | 3,964 |
| F_S_WWC_WM | Fastback | Smooth | Closed | Smooth | Yes | 692 |
| F_S_WWS_WM | Fastback | Smooth | Open | Smooth | Yes | 684 |
| N_S_WW_WM | Notchback | Smooth | Open | Detailed | Yes | 676 |
| N_S_WWC_WM | Notchback | Smooth | Closed | Smooth | Yes | 386 |
| N_S_WWS_WM | Notchback | Smooth | Open | Smooth | Yes | 341 |
| **Total** | | | | | | **8,129** |

Notable points: Fastback + Detailed underbody (F_D_WM_WW) comprises 3,964 samples, approximately 49% of the dataset. Additionally, vehicles with Detailed underbody exist only in the Fastback category; all Estateback and Notchback vehicles have Smooth underbody. This imbalance is the primary cause of performance differences in cross-category generalization experiments (Section 3.4).

## D  SKETCH IMAGE GENERATION PROCEDURE

This section details the sketch image generation procedure described in Section 2.2.

**Generation Pipeline.** Sketch images are generated in three stages:

1. **Normal map rendering**: Generate normal maps from each viewpoint of the 3D mesh. Normal maps encode surface orientation as colors, where differences in orientation between adjacent faces appear clearly as edges, enabling stable edge extraction for line drawing conversion.

2. **Image-to-image translation**: Qwen-Image-Edit-2511 is used for conversion from normal maps to sketch images. Positive prompt: "Line art, Clean outlines, Precise edge detection, Minimalist, Monochrome"; Negative prompt: "shading, shadows, gradients, 3d, messy, extra lines". Seed value is fixed to 7 for all images to ensure reproducibility. Note that this sketch generation is part of data preprocessing, separate from Cp prediction model training and inference.

3. **Post-processing (black line extraction)**: Sketches generated by diffusion models contain subtle noise such as non-white backgrounds. We convert images to grayscale and apply soft mask processing (fade range 30) with brightness threshold 240 to extract black lines while preserving anti-aliasing, obtaining line drawings with white background.

**Why We Use Line Drawing Sketches.** The original DrivAerNet++ dataset does not include sketch images from all viewpoints, necessitating new generation. While colored design sketches (including shading and colors) could be generated via image-to-image translation, we intentionally adopt binary line drawings (black lines on white background). Reasons:

- **Elimination of style variation**: Colored sketches vary in shading style and artistic style across designers, containing high-variance information irrelevant to pressure prediction. Line drawings eliminate these variations, retaining only shape information.

- **Avoiding overfitting to generation artifacts**: Prior work has shown that models trained on diffusion-generated images can overfit to generator-specific artifacts (frequency characteristics, etc.), causing performance degradation (Yamaguchi & Fukuda, 2023; Chu et al., 2025). Line drawing representation minimizes these artifacts.

- **Inductive bias toward geometry**: Discarding color channels constrains the model's hypothesis space to geometric features, encouraging focus on shape information relevant to aerodynamic behavior.

# E    QUALITATIVE RESULTS

This section shows qualitative results for the 8 samples used in the training convergence experiment (Section 3.2.1).

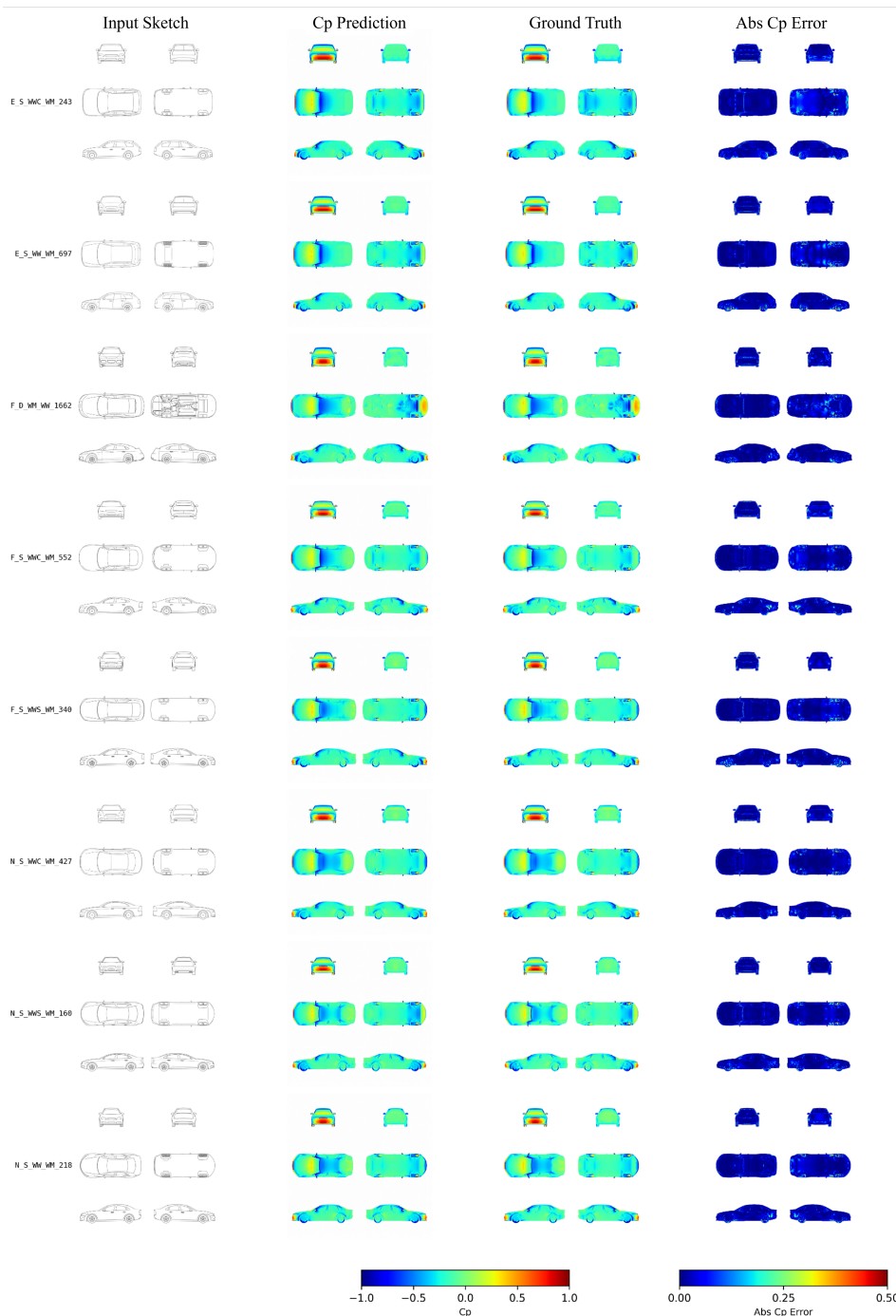

Figure 11: Qualitative results for 8 samples (after ∼29,000 training steps). From left: input sketch, model prediction, CFD ground truth pressure distribution, absolute error map. Includes samples from three vehicle categories (Estateback, Fastback, Notchback). Error maps display white (error 0) to red (high error). Predictions accurately capture global pressure distribution patterns, with errors concentrated primarily at edge regions and fine structural details.

# F    QUALITATIVE ANALYSIS OF DIFFUSION ENSEMBLE

This section qualitatively demonstrates the effects of diffusion ensemble discussed in Section 3.2.2.

Figure 12 shows individual samples (single diffusion samples) from 5 different seeds (seed=0–4) and prediction by their ensemble mean.

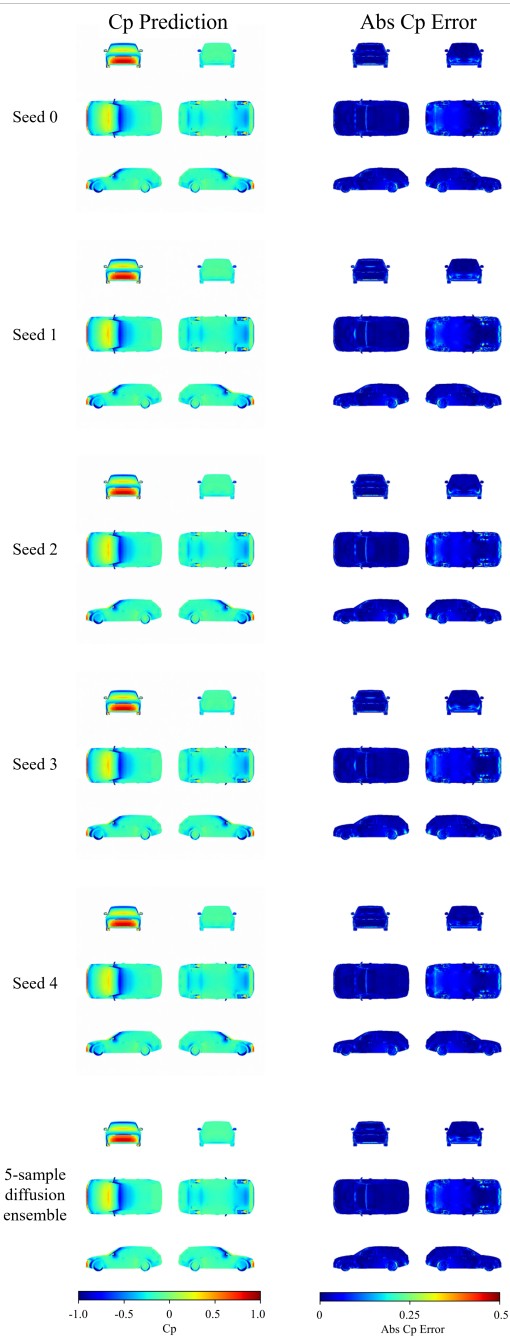

Figure 12: Effect of diffusion ensemble. Left column: predicted Cp distribution, right column: absolute error map. Top 5 rows show individual sample results (seed=0–4), bottom row shows 5-sample diffusion ensemble result. Local variations are visible across individual samples, but ensemble mean reduces variation yielding smoother and more accurate predictions. Error maps confirm that 5-sample ensemble (bottom row) has lower overall error than individual samples.

# G    QUALITATIVE RESULTS FOR DATA EFFICIENCY EXPERIMENT

This section shows qualitative results for the data efficiency experiment discussed in Section 3.3. Figure 13 compares prediction results at $n$=128 and $n$=5,817.

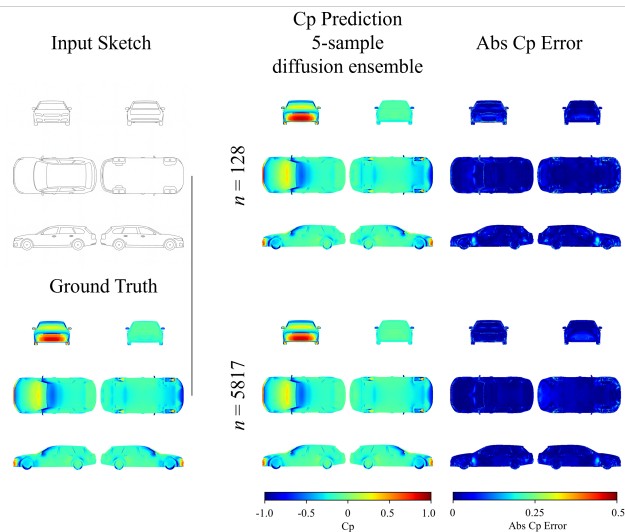

Figure 13: Prediction comparison by training data size (E_S_WWC_WM_323). Top row: $n$=128 prediction, bottom row: $n$=5,817 prediction. Left: input sketch and CFD ground truth, center: predicted Cp distribution (5-sample diffusion ensemble), right: absolute error map.

Despite quantitative differences in Rel L2 of 0.218 ($n$=128) vs. 0.171 ($n$=5,817), visually no notable difference in prediction quality is apparent. Global patterns including front high-pressure region, roof negative pressure, and rear pressure recovery are accurately captured even with limited data ($n$=128). This qualitatively supports that LoRA adaptation of pre-trained models can achieve practical performance with limited data.

# H    QUALITATIVE RESULTS FOR CROSS-CATEGORY PREDICTION

This section shows qualitative results for cross-category prediction discussed in Section 3.4.1. We compare generalization performance to Fastback vehicles by a model trained only on E+N, for F_D and F_S. For F_D (Figure 14), the E+N trained model shows notable errors in underbody and front regions. This is because complex flow patterns specific to Detailed underbody cannot be predicted. Meanwhile, for F_S (Figure 15), no significant difference in error distribution is seen between models, showing good generalization performance for Smooth underbody vehicles even with E+N training. These results visually support the quantitative performance differences shown in Table 5 (F_D: Rel L2 = 0.410 vs. F_S: Rel L2 = 0.186).

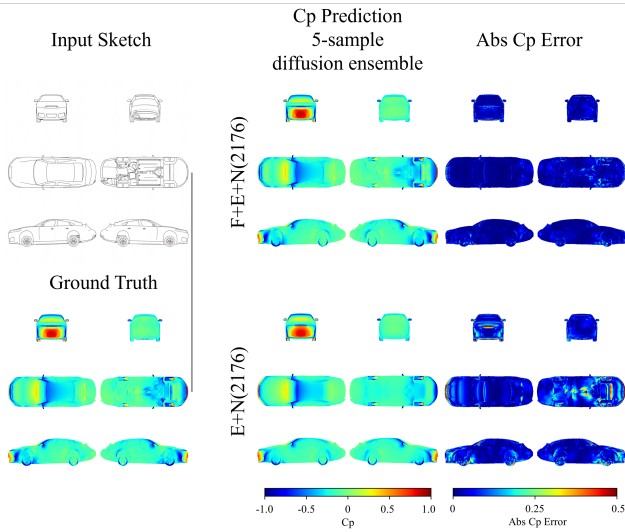

Figure 14: Prediction results for F_D(F_D_WM_WW_3941). Top row: F+E+N trained model, bottom row: E+N trained model.

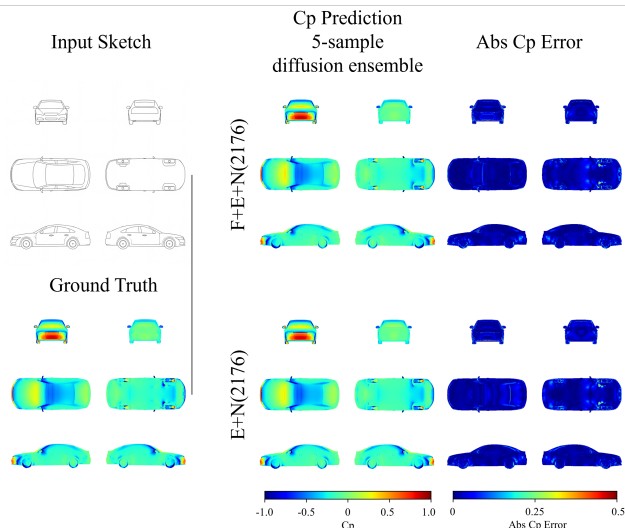

Figure 15: Prediction results for F_S(F_S_WWC_WM_089). Top row: F+E+N trained model, bottom row: E+N trained model.

# I   BASE CATEGORY GENERALIZATION

This section complements Section 3.4.1, verifying base category generalization under limited data conditions.

**Aim.** Section 3.4.1 showed almost no performance difference between F+E+N and E+N with 2,176 training samples. We verify whether differences emerge under limited data conditions.

**Settings.** We compare F+E+N and E+N at 128 samples or fewer, where performance degradation was confirmed in Section 3.3. To eliminate confounding factors, we use subsets with unified tire groove, wheel shape, and underbody configuration (E_S_WWC_WM, F_S_WWC_WM, N_S_WWC_WM).

**Results.** Figure 16 shows training curves for each condition.

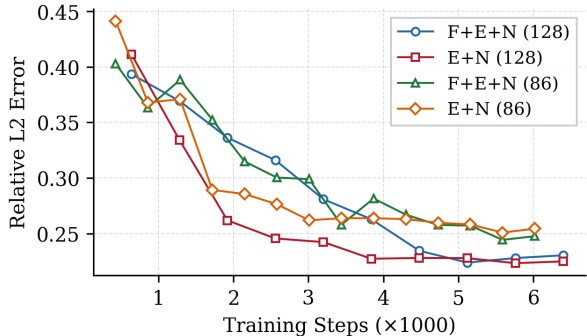

Figure 16: Training curves for base category generalization. Horizontal axis: training steps ($\times 1000$), vertical axis: Rel L2 Error. Evaluation target is F_S_WWC_WM (Fastback).

Table 9: Base category generalization (evaluation target: F_S_WWC_WM, validation set)

| Training categories | Samples | Composition | Rel L2 | $R^2$ |
|---|---|---|---|---|
| F+E+N | 128 | 42:43:43 | 0.220 | 0.921 |
| E+N | 128 | 64:64 | 0.212 | 0.926 |
| F+E+N | 86 | 28:29:29 | 0.239 | 0.906 |
| E+N | 86 | 43:43 | 0.247 | 0.900 |

Final performance is nearly identical for both conditions: at 128 samples, E+N (Rel L2 = 0.212) vs. F+E+N (Rel L2 = 0.220); at 86 samples, E+N (Rel L2 = 0.247) vs. F+E+N (Rel L2 = 0.239). Including Fastback (the evaluation target) in training did not improve performance.

However, examining training curves in Figure 16, differences in convergence speed are visible. E+N conditions (red, green) begin converging approximately 1,000 steps earlier than F+E+N conditions (blue, orange) at 128 samples, with similar trends at 86 samples. Counter-intuitively, E+N, which excludes the evaluation target Fastback, converges faster than F+E+N which includes it. One interpretation is that increasing categories broadens the shape distribution in training data, requiring the model to simultaneously learn more diverse patterns. With the same total sample count, fewer categories mean more samples per category, potentially stabilizing convergence through more learning opportunities for each shape pattern.

## J    APPLICATION TO HAND-DRAWN SKETCHES

**Aim.** Verify that our method functions on hand-drawn sketches actually used in early design stages. Previous experiments used sketches generated via image-to-image translation as described in Section 2.2 (hereafter, AI-generated sketches), but real-world use cases involve human-drawn sketches.

**Settings.** For the same vehicle (E_S_WWC_WM_323), we make predictions using both generated sketches and hand-drawn sketches and compare performance. To enable strict quantitative comparison with CFD ground truth, hand-drawn sketches were created by human tracing of normal map images into 6-view line drawings. While these sketches are not pixel-accurate copies, they include natural variations in line position and thickness introduced by human drawing.

**Results.** Table 10 shows quantitative comparison results.

Table 10: Prediction performance comparison between AI-generated and hand-drawn sketches

| Input Sketch | Rel L2 | $R^2$ | MAE |
|---|---|---|---|
| AI-generated | 0.164 | 0.954 | 0.0252 |
| Hand-drawn | 0.219 | 0.920 | 0.0223 |

Figure 17 shows qualitative comparison results.

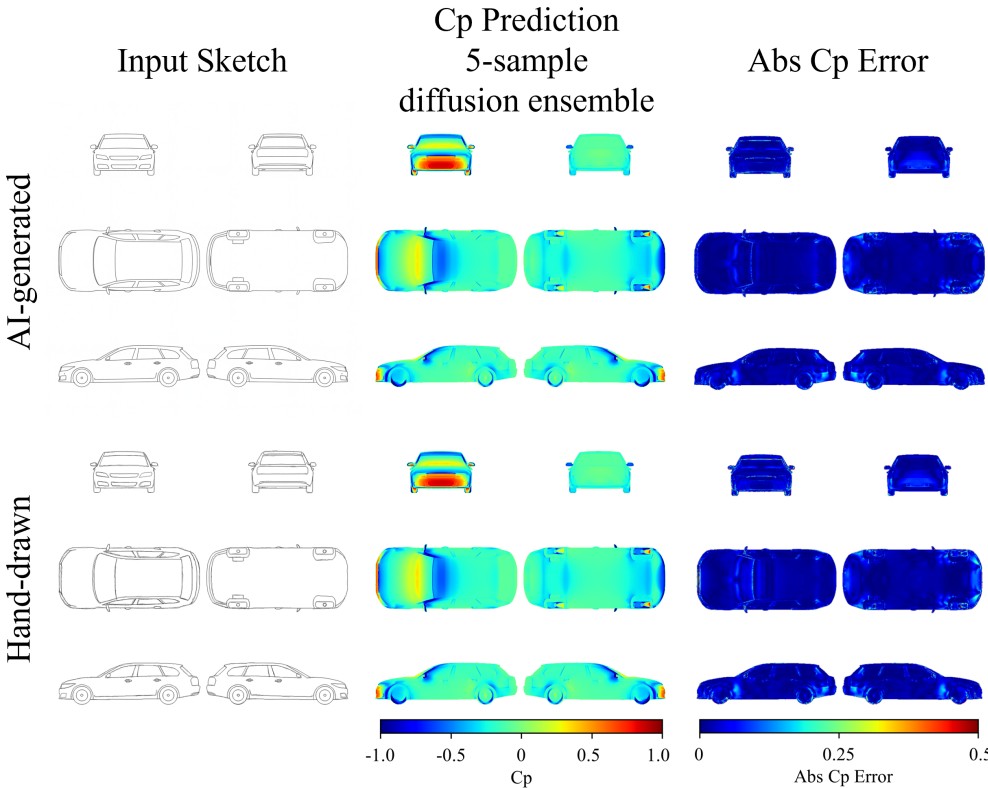

Figure 17: Prediction comparison between AI-generated and hand-drawn sketches. Top row: prediction from AI-generated sketch. Bottom row: prediction from hand-drawn sketch. From left: input sketch, predicted Cp distribution, CFD ground truth, absolute error map.

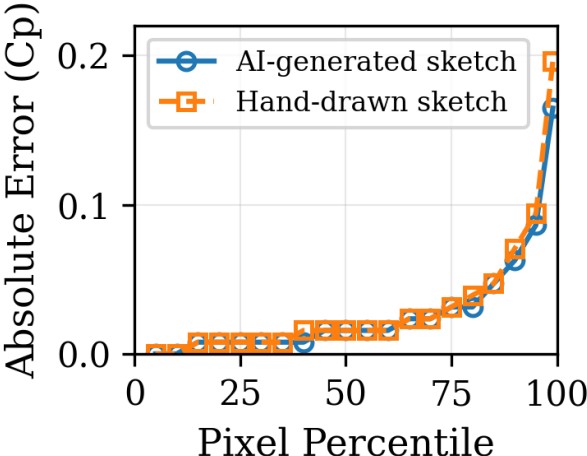

Figure 18: Error distribution comparison between AI-generated and hand-drawn sketches. Horizontal axis: percentile (pixels sorted by ascending error), vertical axis: absolute error.

Hand-drawn sketches achieve Rel L2 = 0.219, somewhat worse than AI-generated sketches (Rel L2 = 0.164), but global pressure distribution patterns are accurately captured. Figure 18 shows error distribution comparison.

Error distribution analysis shows nearly equivalent accuracy for both up to the 75th percentile, with differences emerging primarily in high-error regions (80th percentile and above). Human tracing naturally introduces variation in line position and thickness, making mechanical precision in contour reproduction difficult, and local error increases are unavoidable. However, global shape information is sufficiently conveyed, achieving practical accuracy of Rel L2 = 0.219 even with hand-drawn sketches. These results demonstrate that "aerodynamic trend assessment at early design stages without 3D CAD," the main objective of this work, is achievable using actual human-drawn sketches.

# K QUALITATIVE COMPARISON OF STANDARD DEVIATION AND ERROR MAPS

This section qualitatively demonstrates the correspondence between ensemble variance and prediction error discussed in Section 4. We compare standard deviation maps across 5 seeds with absolute error maps between mean prediction and CFD ground truth.

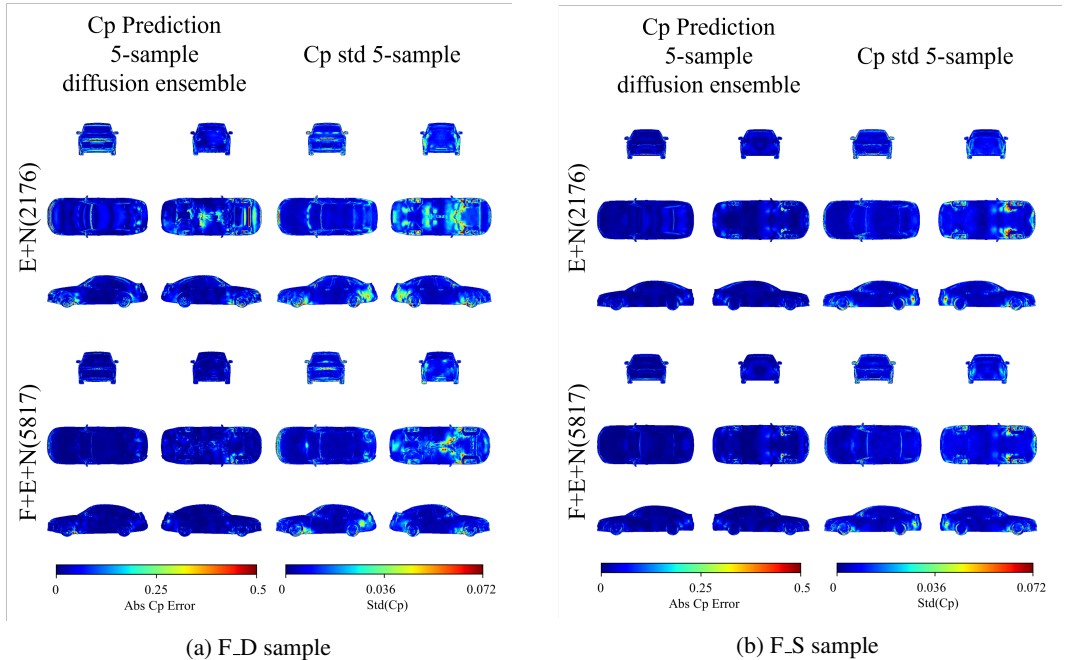

(a) F_D sample                                         (b) F_S sample

Figure 19: Comparison of standard deviation and error maps. Left column: std. dev. map, right column: error map. (a) F_D: E+N model shows high error in underbody but low std. dev. ("overconfident"). (b) F_S: High std. dev. regions correspond to high error regions for both models.

In E+N training evaluated on F_D (Figure 19a), the model makes consistent (but incorrect) predictions for flow patterns specific to Detailed underbody, with variance not reflecting error. This can be interpreted as "overconfident" prediction for shapes absent from training data. Meanwhile, for F_S (Figure 19b), good spatial correspondence between standard deviation and error is observed for both training conditions, confirming that variance functions as a confidence indicator.

