# OpenReview forum: "Vehicle Surface Pressure Prediction from 2D Sketches via a Pre-Trained Diffusion Model"
_ICLR.cc/2026/Workshop/FM4Science — ICLR 2026 Workshop FM4Science Poster_

### Official Review · Reviewer_YJh8 · 2026-02-24
**Interesting work!**

**Rating:** 7
**Confidence:** 5

**Review:**

This paper presents an interesting application of pre-trained diffusion models to aerodynamic surface pressure prediction from 2D vehicle sketches. While the methodological novelty is limited — as the core technical components (diffusion models, LoRA fine-tuning, ensemble sampling) are established techniques — the contribution lies in their integration within a new and practically meaningful application domain. The problem is well-motivated, particularly for early-stage automotive design where 3D geometry is unavailable, and the experimental study is systematic and carefully analyzed.

The experimental design is systematic and thorough, including careful exploration of inference parameters, data efficiency, cross-category generalization, and uncertainty behavior. One limitation is the absence of a non-diffusion baseline such as a U-Net model for comparison. Including a simpler baseline would help clarify whether the gains stem from the modelling choices or rather from the dataset and problem formulation.

---

### Official Review · Reviewer_2AUD · 2026-02-24
**A carefully executed and practically motivated study with limited methodological novelty**

**Rating:** 6
**Confidence:** 3

**Review:**

This paper proposes predicting surface pressure coefficient (Cp) distributions directly from 2D vehicle sketches using a pretrained diffusion model (Qwen-Image-Edit) fine-tuned with LoRA. The goal is to enable aerodynamic evaluation at the sketch stage, before 3D CAD models are available.

Overall, the work is practically motivated and carefully executed. The motivation is reasonable and clearly presented. However, the methodological contribution is limited as there are no new modeling techniques introduced, and the approach mainly applies an existing pretrained diffusion framework to a new application domain. My main concerns relate to missing baselines and incomplete evaluation of some key claims.

Pros:
1. Clear and practical motivation. The idea of estimating aerodynamic surface pressure directly from sketches is interesting and relevant to early-stage vehicle design. The problem setting is well explained and makes sense from an engineering perspective.
2. Well-structured and readable. This paper is clearly written and easy to follow. Each section has a clear purpose, and the experimental setup is described in a structured way. The figures and tables help illustrate the results effectively.
3. Systematic experimental design. The authors explore diffusion inference parameters, LoRA rank and alpha, ensemble averaging, and data efficiency. The underbody configuration experiment is helpful to show where generalization breaks down.

Cons:
1. Limited methodological novelty. The core technical components (diffusion + LoRA) are existing methods. The contribution lies primarily in applying them to a new problem rather than advancing modeling techniques.
2. Lack of baselines. The paper is rich in self-parameter comparisons (e.g., LoRA rank, inference steps, ensemble size) but lacks cross-model comparisons. There is no evaluation against alternative architectures such as UNet-style image-to-image regression or other deterministic models. As a result, it is unclear how much this task truly benefits from a very large (20B parameter) diffusion model. The claimed improvement from diffusion ensemble averaging would be more convincing if compared against simpler models.
3. Missing computational cost report. Training time, inference latency, GPU memory requirements, and storage footprint are not reported, which is important for understanding the applicability of the proposed method. Also, adding similar details for the baseline models (see point 2) would be useful to show whether the large pretrained diffusion models are a good fit for this task.
4. Uncertainty is not calibrated. The paper reports a correlation between ensemble variance and error, but correlation alone does not establish calibration. No reliability or coverage analysis is provided, so the claim of “calibrated uncertainty” appears overstated.

---

### Official Review · Reviewer_PHVp · 2026-02-25
**Sketch to surface pressure via fine-tuned diffusion shows promise with some OOD limitations.**

**Rating:** 7
**Confidence:** 3

**Review:**

### Summary

This paper proposes an end-to-end framework for predicting vehicle surface pressure coefficient (Cp) distributions directly from 2D sketch inputs, bypassing 3D geometry reconstruction and CFD simulation. The method fine-tunes an image-editing diffusion model (Qwen-Image-Edit-2511) using LoRA and treats Cp prediction as an image-to-image translation task. On DrivAerNet++, the approach achieves strong performance, maintains usable accuracy down to 128 training samples, and shows that generalization depends more on geometric feature coverage than on coarse category labels. Diffusion ensemble inference improves accuracy and provides a partial uncertainty signal, though this degrades under structural OOD conditions.

### Evaluation

The experimental structure is systematic and well executed, covering feasibility, hyperparameter sensitivity, data efficiency, generalization, and uncertainty analysis. The writing is clear, with well supported claims and detailed appendices. The individual components are not new, but the application of a large pre-trained diffusion model to sketch-to-Cp surrogate modeling is novel and thoughtfully analyzed. The insights on geometric feature coverage and ensemble behavior are particularly valuable. The work has good practical relevance for aerodynamic design by removing the need for explicit 3D simulation. However, its impact depends on generalization beyond the specific dataset and CFD setup studied. Results show degradation in performance and in UQ reliability on OOD structures. In the data efficiency study, it seems like the models haven’t fully converged, and the curves suggest that the gap between SOTA and low-data may grow when trained for longer.

### Pros

- Clear and practically motivated problem with good relevance to early stage design.
- Strong quantitative performance on the studied dataset.
- Insightful generalization study identifying feature-coverage limits.
- Meaningful low-data analysis demonstrating robustness with limited training samples.
- Insightful analysis of diffusion ensemble bias/variance behavior.
- Honest discussion of uncertainty limitations.
- Well written with detailed appendices.

### Cons

- Evaluation restricted to a single dataset and fixed CFD setup; external validity unclear.
- Data-efficiency study may not fully control for convergence differences.
- Ensemble-based uncertainty degrades under OOD geometry.
- Full impact on design workflow is not described in detail. Results not fully put into that context.

---

### Official Review · Reviewer_A2ca · 2026-02-25
**Review of Vehicle Surface Pressure Prediction from 2D Sketches via a Pre-Trained Diffusion Model**

**Rating:** 7
**Confidence:** 4

**Review:**

The paper proposes an end-to-end approach to direct predict surface pressure coefficient (Cp) distributions from 2D vehicle sketch images via image-to-image translation to allow for aerodynamic feedback at the earliest design stage. The work leverages LoRA adaptation of a pre-trained diffusion model (Qwen-Image-Edit-2511) and improves prediction accuracy via diffusion ensemble. Prior work, focuses on predicting directly from 3D geometry, which cannot be applied at the sketch stage or route through 3D geometry from 2D geometry, but the evaluated geometry is often not as precise.  Practical accuracy is maintained even with limited data, and the work enables the feasibility of aerodynamic evaluation at the earliest stage (sketch).

**Strengths:**
* Enables aerodynamic guidance directly from sketches, potentially shortening design loops before CAD/3D geometry is available, is a compelling contribution.
* The experimental order design is detailed and shows the rigor involved to obtain the ideal set of LoRA hyperparameters, inference parameters, and complemented by the LoRA Hyperparameter Sensitivity analysis.
* The subset evaluation, validity verification, and data efficiency experiments are well-motivated and show the efficiency of the approach.
* The bias-variance tradeoff analysis of steps=3 vs steps=10 in the context of diffusion ensemble (Section 3.2.2) is insightful and well-grounded in the MSE decomposition framework. This provides useful practical guidance for practitioners applying diffusion models to physics prediction.
* The uncertainty quantification discussion (Section 4) is forthcoming, acknowledging that ensemble variance becomes unreliable for out-of-distribution shapes rather than overclaiming UQ reliability.
* The hand-drawn sketch experiment (Appendix J) is a valuable practical validation that bridges the gap between the synthetic training setup and real-world usage.



**Weaknesses**
* The paper exclusively uses Qwen-Image-Edit as single model without comparing against any alternative image-to-image methods. Without baselines, it is unclear how much of the performance comes from the diffusion model paradigm vs. the specific architecture, or whether a simpler/faster model could achieve comparable results. The authors acknowledge that direct comparison with prior work is difficult due to differing modalities, but a non-diffusion image-to-image baseline trained on the same data would strengthen the core contribution.
* There is no end-to-end baseline for the “3D route”, even at reduced scale, to contextualize the direct-2D approach’s trade-offs.
* Evaluation is entirely image-space, not on 3D meshes. The Cp predictions are evaluated as 2D image reconstructions via inverse jet colormap. The evaluation never maps predictions back to 3D geometry. This makes it difficult to assess whether the predictions are useful for downstream engineering tasks (e.g., drag coefficient estimation).
* The sketch generation pipeline might introduce a confound. The sketches are generated using the same Qwen-Image-Edit model used for Cp prediction (just with different prompts). This creates a potential data leakage concern the prediction model may be exploiting artifacts of its own generation process rather than learning true sketch→pressure mappings.



**Clarification**
* How are the six views arranged in the 1024×1024 composite image? Is the layout fixed, and does the spatial arrangement affect prediction quality?



**Overall Decision: 7/10:** The paper addresses a well-motivated practical gap, aerodynamic evaluation at the sketch stage, and demonstrates feasibility convincingly. The experimental methodology is thorough, and the analysis of generalization, data efficiency, and uncertainty is robust. However, the lack of any baseline comparison is a significant gap that makes it hard to assess the marginal contribution of the specific technical choices. The evaluation remaining entirely in image space (rather than mapping back to 3D surface quantities) also limits the engineering utility claims. With baselines and a more grounded evaluation, this could be an even stronger contribution.

---

### Official Review · Reviewer_ZrzH · 2026-02-25
**Promising Application Paper with Minor Clarifications Suggested**

**Rating:** 6
**Confidence:** 4

**Review:**

Overall assessment:
Well-written paper addressing an opportunity of 3D aerodynamic prediction from the 2D sketches. The approach is interesting and relevant. I have several comments that could help further strengthen the manuscript:

Typos:
Figures 3, 5, and 13: “grand truth” → “ground truth”.


Figure 2 (Page 4):
Please add a description of the error bars shown in the figure (e.g., what they represent and how they were computed).


Ensemble justification (Figures 5 and 12, Appendix F):
The current figures do not clearly demonstrate that the ensemble mean reduces prediction variation and improves accuracy. Consider adding individual color bars for each row (or explicitly reporting value ranges) to better illustrate how Cp and absolute error values change because of the effect of diffusion ensemble.

CFG selection (Page 6):
The statement “we select the widely-used 4.0 for reproducibility” would benefit from supporting references.

Scope and practical implications:
The manuscript suggests that Cp distributions can be predicted directly from 2D sketches. Could the authors clarify whether this approach eliminates the need for full 3D geometric representations in downstream aerodynamic analysis, or whether it is intended primarily as an early-stage design screening tool prior to 3D simulation?
It would also be helpful to discuss practical implications within the automotive design pipeline, such as potential reductions in computational cost, design iteration time, etc.

---

### Meta-Review · Area_Chair_MPG4 · 2026-02-27

**Recommendation:** Accept (Poster)
**Confidence:** 4

**Metareview:**

The average review score is above 6, which means reviewers recommended an acceptance.

---

### Decision · Program_Chairs · 2026-03-03

Accept (Poster)